# Mepolizumab Improves Outcomes of Chronic Rhinosinusitis with Nasal Polyps in Severe Asthmatic Patients: A Multicentric Real-Life Study

**DOI:** 10.3390/jpm12081304

**Published:** 2022-08-10

**Authors:** Stefania Gallo, Paolo Castelnuovo, Luca Spirito, Marta Feduzi, Veronica Seccia, Dina Visca, Antonio Spanevello, Erica Statuti, Manuela Latorre, Claudio Montuori, Angela Rizzi, Cristina Boccabella, Matteo Bonini, Eugenio De Corso

**Affiliations:** 1Department of Otorhinolaryngology, Ospedale di Circolo e Fondazione Macchi, ASST Sette Laghi, 21100 Varese, Italy; 2UPLOAD (Upper and Lower Airways Inflammatory Diseases) Research Center, University of Insubria, 21100 Varese, Italy; 3Otolaryngology Audiology and Phoniatric Operative Unit, Department of Surgical, Medical, Molecular Pathology and Critical Care Medicine, Azienda Ospedaliero Universitaria Pisana, University of Pisa, 56124 Pisa, Italy; 4Division of Pulmunary Rehabilitation, Istituti Clinici Scientifici Maugeri, IRCCS, 21049 Tradate, Italy; 5Department of Medical Specialties, Pulmonary Unit, Hospital of Massa, 54100 Massa, Italy; 6Department of Head and Neck and Sensory Organs, Catholic University of the Sacred Heart, 00168 Rome, Italy; 7Unit of Allergology, A. Gemelli Hospital Foundation IRCCS, 00168 Rome, Italy; 8Department of Internal Medicine and Geriatrics, Catholic University of the Sacred Hearth, 00168 Rome, Italy; 9Unit of Pulmonology, A. Gemelli Hospital Foundation IRCCS, 00168 Rome, Italy; 10Unit of Otorhinolaryngology, A. Gemelli University Hospital Foundation IRCCS, 00168 Rome, Italy

**Keywords:** chronic rhinosinusitis, nasal polyps, severe asthma, mepolizumab, SNOT-22, nasal polyp score, eosinophil, IL-5, biologics

## Abstract

Objective: The upcoming introduction of mepolizumab represents a promising treatment for chronic rhinosinusitis with nasal polyps (CRSwNP). The present study aimed to evaluate the effectiveness of mepolizumab on sinonasal outcomes of comorbid CRSwNP and severe asthma in a real-life setting. The primary endpoint was to evaluate changes in the SinoNasal Outcome Test (SNOT)-22 score, Nasal Polyp (NP) score, and blood eosinophil count during a 12-month treatment with mepolizumab. Secondary endpoints were to quantify mepolizumab’s effects on the mentioned parameters, identify clinical variables influencing the degree of response to treatment, and portray responder and nonresponder patients. Methods: A multicentric retrospective no-profit observational study on severe asthmatic patients, treated with mepolizumab, and comorbid CRSwNP was conducted. All patients were followed for at least 12 months. SNOT-22 score, NP score, and blood eosinophil count (and other CRS-specific variables) were collected at baseline and after 12 months. Results: Forty-three patients were included. A statistically significant reduction was observed for SNOT-22 score (mean t0 SNOT-22 54.8 ± 25.9; mean t12 SNOT-22 31.5 ± 21.3, *p* < 0.0001), NP score (median t0 NPS 3 (IQR 3); median t12 NPS 2 (IQR 4), *p* < 0.0001), and blood eosinophil count (mean t0 blood eosinophils 804.7 ± 461.5 cell/µL; mean t12 blood eosinophils 107.5 ± 104.6 cell/µL, *p* < 0.0001) after 12 months of treatment. Twenty patients (47%) gained improvement both in clinical and endoscopic outcome. Mepolizumab responder patients presented a t0 SNOT-22 significantly higher than nonresponders (*p* = 0.0011). Conclusions: Mepolizumab improved CRSwNP outcomes in a population of severe asthmatic patients. No clinical feature emerged to outline the profile of a “typical” responder patient, except for baseline SNOT-22 score, which seemed to affect the response to treatment. Further studies would be necessary to supplement these preliminary evaluations.

## 1. Introduction

Mepolizumab (SB-240563) is a fully humanized IgG1/k-class monoclonal antibody that selectively binds to interleukin-5 (IL-5), preventing its engagement with IL-5 receptor alpha (IL-5Rα) and finally inhibiting its downstream activities. Because of its high-affinity interaction, mepolizumab does not appear to interfere with other cytokines and exhibits a good safety and tolerability profile [1]. IL-5 is implicated in a wide range of biological functions of eosinophils, which are well-known players in airways’ inflammation [2].

A regimen of 100 mg of subcutaneous mepolizumab, administered once every 4 weeks, has been approved since 2015 for the add-on maintenance treatment of patients over 6 years old affected by moderate-to-severe eosinophilic asthma (SEA), being uncontrolled despite maximal standard therapy according to the Global Initiative for Asthma (GINA) guidelines [3]. In addition, mepolizumab recently gained approval by the FDA (Food and Drug Administration) and EMA (European Medicines Agency) for use also in hypereosinophilic syndrome, eosinophilic granulomatosis with polyangiitis (EGPA), and chronic rhinosinusitis with nasal polyps (CRSwNP).

From previous trials, mepolizumab proved to be highly effective in treating SEA. The benefits on the lower airways included a reduction in exacerbations’ rates and in asthma control questionnaires scores (asthma control questionnaire, ACQ; asthma control test, ACT) [4] and an improvement in asthma symptoms and quality of life (QoL). This clinical relief was paralleled by favorable changes in airflow limitation (FEV1) [5] and overall lung function [6]. Moreover, post hoc analysis of the mentioned studies showed that patients with SEA obtained clinical improvement regardless of CRSwNP status, but benefits were achieved to a greater extent in patients with nasal polyps (NP) than in patients without NP [7].

On the other hand, knowledge on the effects of mepolizumab on CRSwNP in patients with SEA is not yet conclusive. The SYNAPSE trial reported how mepolizumab is effective at multiple levels, in decreasing the occurrence of nasal surgeries and the use of oral corticosteroids (OCS) and in improving nasal symptoms [8]. In addition, mepolizumab reduced the nasal polyp score (NP score) and nasal obstruction visual analog score (VAS) regardless of comorbid asthma. Other studies on the effect of mepolizumab on asthma investigated the effect on CRSwNP as a secondary endpoint. However, analyses often rely only on patient-reported outcomes (sometimes not even standardized with validated scoring systems) [9]. Moreover, some discrepancies on the ability of mepolizumab to reduce NP volume emerged from isolated real-life studies [4,10,11].

Considering these premises, the aim of this study was to verify the efficacy of mepolizumab on sinonasal aspects in a specific subset of patients affected by SEA and concomitant CRSwNP in real life. The primary endpoint was to evaluate changes in the SinoNasal Outcome Test 22 (SNOT-22), NP score, and blood eosinophil count during a 12-month treatment with mepolizumab. Secondary endpoints were to quantify mepolizumab’s effects on the mentioned parameters, identify clinical variables influencing the degree of response to treatment, and portray responder and nonresponder patients.

## 2. Materials and Methods

### 2.1. Study Population

A retrospective no-profit observational multicentric study was conducted revising clinical data from adult patients affected by SEA and concomitant CRSwNP who started mepolizumab treatment according to GINA guidelines [3,12,13,14,15] between July 2017 and September 2020 for whom data at baseline and after 12 months of biological therapy were available. The analyses were concluded in November 2021. Some of the included patients had already been included in a previous paper by the same authors [11].

Existing diagnosis of CRSwNP was verified in accordance with the European Position Paper on Rhinosinusitis and Nasal Polyps (EPOS) criteria [16]. Patients affected by secondary CRS were excluded a priori. It should be noted that as mepolizumab was, at that time, licensed only for the treatment of SEA, and not yet for CRSwNP, the choice of biological treatment depended exclusively on the features of asthma. It is also true that, prior to treatment, all included patients presented a blood eosinophil count ≥250 cells/mm^3^ and/or peripheral total IgE ≥ 100 kU/L, which are EPOS indicators of a “type 2 inflammation” profile of the airways [16].

The study was conducted in compliance with the Helsinki Declaration and with policies approved by the Insubria Board of Ethics. Informed consent was obtained from all participants included in the study.

### 2.2. Study Design

Baseline data were collected, concerning demographic features, smoking habits, chronic rhinosinusitis (CRS) and asthma onset, non-steroidal anti-inflammatory drug (NSAID) intolerance, sensitization to common inhalants, type of ongoing nasal therapies, need for OCS, history of previous biological treatment, history, type, and timing of sinonasal surgeries.

A set of five clinical parameters, acquired at baseline (t0) and at 12 months of treatment (t12), was examined to accomplish the study endpoints: SNOT-22 score, approximated overall CRS symptoms, and CRS-related QoL; scores defined by the first 12 items of SNOT-22 (SNOT 1-12) and by some individual symptoms (nasal obstruction, nasal discharge, loss of smell, headache, and ear fullness) were also evaluated; NP score expressed the endoscopic extension of NP; blood eosinophil count was assumed as a surrogate of eosinophilic inflammation of the airways; the ACT score and EPOS assessment scale were employed to evaluate, respectively, asthma and CRS control of disease over 12 months of observation. The EPOS scale for CRS control is a multimodal scale coupled with standard subjective and objective outcomes of CRS (i.e., nasal obstruction, nasal discharge, facial pain, loss of smell, and endoscopic evidence of diseased mucosa) to items such as sleep disturbance or fatigue and need of rescue (steroid or antibiotic) treatment for disease control. The advancement of at least one step on the clinical control scale (for both ACT and EPOS scales) was classified as *improvement*, respectively, for asthma and CRSwNP.

The magnitude of mepolizumab effect was first defined based on the size of changes (delta, Δ) of the clinical assessment parameters (SNOT-22 score, NP score, and blood eosinophil count) between t0 and t12.

*Clinical improvement* was defined on achieving at least 1 MCID (Minimal Clinical Important Difference) in SNOT-22 score. The MCID is the lowest degree of change that a patient will notice, which, for SNOT-22 score, has previously been defined as 8.9 points [17]. *Endoscopic improvement* was defined on achieving at least a 1-point reduction in NP score. Patients who achieved an improvement both in SNOT-22 score and NP score were classified as *overall responder*; those who obtained benefits either on clinical or endoscopic scores were considered as *partly responder* and those who did not improve in any aspect as *nonresponder*.

All adverse events potentially related to the use of the drug during the observation period were collected.

### 2.3. Statistical Analysis

The distribution of continuous variables was verified using the Kolmogorov–Smirnov test. The distributions of age, SNOT-22, and blood eosinophils values were found normal (*p* > 0.1), whereas the distributions of the other variables were not normal. Analyses were performed with parametric tests for normally distributed variables (Student’s t-test and One-way ANOVA) and nonparametric tests for nonnormally distributed variables, and ordinal and nominal variables (Wilcoxon test, Mann–Whitney test, Kruskal–Wallis test, Fisher’s exact test, and Yates’s chi-squared test). Results are reported as mean ± standard deviation (SD) for normally distributed variables and median with interquartile range (IQR) for nonnormally distributed variables.

Statistical analysis was performed using GraphPad Prism Version 9.2.0 (GraphPad Software, Inc., San Diego, CA, USA) and *p*-value < 0.05 was considered significant.

## 3. Results

### 3.1. Baseline Characteristics of the Population

A total of 45 severe asthmatic patients with comorbid CRSwNP treated with mepolizumab were identified. Two subjects did not conclude the 12-month observation period, due to early discontinuation of treatment (4 months) because of disabling adverse events (AEs). As a result, subsequent analyses were performed on the remaining 43 patients.

The final population included 18 males and 25 females, with an average age of 52.2 ± 10.9 years. Demographic and clinical data are reported in Table 1.

At baseline, 19 patients (44%) and 24 patients (56%) presented, respectively, with partly controlled asthma and with uncontrolled asthma, based on ACT score. None of the patients had clinically controlled asthma. Regarding CRSwNP, 1 patient (2%) presented with controlled disease, 9 patients (21%) with partly controlled disease, and 33 patients (77%) with uncontrolled disease, according to the EPOS assessment scale. As a result, 17 patients (40%) presented with a concordant uncontrolled clinical status of asthma and CRSwNP, whereas 16 patients (37%) presented with a clinically partly controlled asthma and uncontrolled CRSwNP. These were the two most frequently observed combinations of clinical airway control.

### 3.2. Effects of Mepolizumab on Clinical Parameters

Mean baseline (t0) SNOT-22 score was 54.8 ± 25.9 and mean t0 SNOT 1–12 score was 34.9 ± 13.7. Median t0 scores of SNOT-22-specific symptoms were: nasal obstruction 4 (IQR 2), nasal discharge 4 (IQR 3), loss of smell 5 (IQR 2), facial pain 2 (IQR 0), and ear fullness 3 (IQR 3). Median t0 NP score was 3 (IQR 3) and mean t0 blood eosinophil count was 804.7 ± 461.5 cell/µL.

Mean t12 SNOT-22 score was 31.5 ± 21.3 and mean t12 SNOT 1–12 score was 20.4 ± 12.5. Median t12 scores of SNOT-22 individual symptoms were: nasal obstruction 2 (IQR 3), nasal discharge 2 (IQR 4), loss of smell 2 (IQR 5), facial pain 0 (IQR 2.5), and ear fullness 1 (IQR 2). Median t12 NP score was 2 (IQR 4) and mean t12 blood eosinophil count was 107.5 ± 104.6 cell/µL.

A statistically significant difference was observed for all clinical parameters examined at t0 and t12 and, therefore, attributable to the effect of mepolizumab treatment: SNOT-22 score (*p* < 0.0001), SNOT 1-12 score (*p* < 0.0001), NP score (*p* < 0.0001), and blood eosinophil count (*p* < 0.0001) (Figure 1). The same was valid for each individual symptom (nasal obstruction *p* < 0.0001, nasal discharge *p* < 0.0001, loss of smell *p* < 0.0001, facial pain *p* < 0.0001, and ear fullness *p* < 0.0001).

As in-treatment nasal surgery could have represented a bias for the measurement of mepolizumab effect, the same analysis was conducted in the subgroup of the population excluding patients operated during treatment, and the above-mentioned statistically significant differences were fully confirmed.

Concerning *improvement*, 28 patients (64%) improved their asthma status, and 22 patients (51%) their CRSwNP status, suggesting that mepolizumab positively impacts on asthma and the CRSwNP clinical control in more than half the population.

### 3.3. Quantification of Mepolizumab Effects

Median *delta* (Δ) SNOT-22 score was −22 (IQR 31), median *delta* (Δ) NP score was −1 (IQR 2.5), and mean *delta* (Δ) blood eosinophils were −699.9 ± 464.5 cell/mm^3^.

The greatest improvement in SNOT-22 score was obtained by patients with t0 SNOT-22 score ≥40 (*p* = 0.0022), whereas the greatest improvement in NP score was gained by allergic patients (*p* = 0.0412) and patients operated during treatment (*p* = 0.0012). Lastly, the greatest reduction in blood eosinophil count was observed in patients presenting with a t0 ACT uncontrolled asthma (*p* = 0.0374).

Twenty-six patients (60%) were classified as *clinical responders*, achieving at least 1 MCID; the remaining 17 patients (40%) had unchanged (13/17) or worsened (4/17) their clinical status. Among *clinical responders*, the median number of achieved MCID was 3 (IQR 3) and the percentage of relative improvement was −61.2 ± 24.6. There were no statistically significant differences in the distribution of several t0 clinical variables (NSAID sensitivity, allergy, smoking habits, OCS intake, history of surgery, clinical asthma status, NP score, and blood eosinophil count) between *clinical responders* and *nonresponders*, except for baseline SNOT-22 score; mepolizumab *clinical responder* presented a t0 SNOT-22 significantly higher than that of *nonresponders* (*p* = 0.0015).

Twenty-seven patients (63%) were classified as *endoscopic responders*, having experienced a reduction of at least 1 point in NP score; the remaining 16 patients (37%) had unchanged (13/16) or worsened (3/16) their endoscopic status. Among *endoscopic responders*, the median of NP score reduction was 2 (IQR 1). There were no statistically significant differences in the distribution of several t0 clinical variables (NSAID sensitivity, smoking habits, OCS intake, history of surgery, clinical asthma status, SNOT-22 score, NP score, and blood eosinophil count) between *endoscopic responders* and *nonresponders*, except for atopic status; allergic patients are, indeed, significantly more represented among mepolizumab *endoscopic responder* than *nonresponders* (*p* = 0.0293).

Combining both clinical and endoscopic effects, 20 patients (47%) were classified as *overall responder* and 23 patients (53%) as *partly* (13/23) or *nonresponder* (10/23). There were no statistically significant differences in the distribution of several t0 clinical variables (NSAID sensitivity, allergy, smoking habits, OCS intake, history of surgery, clinical asthma status, NP score, and blood eosinophil count) between *overall responders* and *nonresponders*, except for baseline SNOT-22 score; mepolizumab *overall responders* presented a t0 SNOT-22 significantly higher than *nonresponders* (*p* = 0.0011).

### 3.4. Tolerability Profile

Apart from the 2 patients (4.4%) whose biologic treatment was discontinued due to AEs (arthritis in one case and gastroenteritis in the other), no other serious AEs were reported in the remaining 43 patients. However, 11 patients (25.6%) reported one or more minor AEs, including nasopharyngitis (4), pharyngodynia (4), arthralgia (3), flu (3), back pain (1), stomachache (1), pyrexia (1), and headache (1).

## 4. Discussion

Patients affected by SEA frequently present with comorbid CRSwNP [7]. It is estimated that more than 40% of severe asthmatics suffer from CRSwNP [18]; this percentage even increases to 60% when considering late-onset SEA [19]. CRSwNP is known to impact asthma severity and, for this reason, it is included among asthma-treatable traits [20]. In eosinophilic asthma, eosinophils, under the action of several cytokines including IL-5, accumulate within the bronchial tract where they release cytotoxic proteins, lipid mediators, cytokines, and chemokines that significantly contribute to airway hyperresponsiveness, inflammation, remodeling, and clinical exacerbations [21]. Similarly, in CRSwNP, eosinophils interfere with mucosal inflammation and remodeling via cytotoxic protein degranulation [22]. Moreover, their derived mediators can damage epithelial cells, stimulate epithelial-to-mesenchymal transition, activate or suppress sensory nerves, and modulate the activity of stem cells and plasma cells [23,24]. The presence of tissue eosinophilia in CRSwNP is frequently associated with extensive sinus disease, higher postoperative symptom scores, less improvement in both disease-specific and general QoL, and a higher polyp recurrence rate [25].

As anti-IL-5 biological agents were introduced as a new-line treatment for patients affected by SEA, the finding of a concomitant sinonasal therapeutic effect in patients with comorbid CRSwNP has sparked interest in the application of such drugs as a possible breakthrough treatment in patients with type 2 eosinophilic difficult-to-treat nasal polyposis.

The effect of mepolizumab on CRSwNP has been evaluated in two phase II randomized clinical trials (RCT), which showed a reduction in NP score from baseline (mean change −1.30 ± 1.72) [26] and, accordingly, both a SNOT-22 score lowering from baseline (51.5 ± 17.0) to week 25 (28.8 ± 22.0) and a need for surgery rate reduction [27]. These findings were reinforced by results from the phase III RCT (SYNAPSE). The administration of 100 mg of subcutaneous mepolizumab once every 4 weeks corroborated the reduction with respect to a baseline of NP score (−0.9 ± 1.90) and SNOT-22 score (−29.4 ± 24.67), the improvement of nasal obstruction VAS score, and further confirmed the reduced need of sinonasal surgery during the 52-week treatment period [8].

The encouraging results obtained in RCTs prompted several authors to evaluate the effect of mepolizumab on CRSwNP in severe asthmatic patients in the real-life setting. A limited number of dissimilar real-life studies (RLS) have been published so far (Table 2).

In line with RLSs results [5,11,30,31] and those reported by RCTs, our study showed a downward trend of NP score compared to baseline (median t0 NPS 3 (IQR 3) and a median t12 NPS 2 (IQR 4), *p* < 0.0001). The *delta* NP score was −1 points (IQR 2.5) and substantially overlapped with that from the SYNAPSE study (−0.8 points) [32]. To date, the NP score is the parameter that provided the most discordant results. Among the RLSs, in which the NP score was calculated, only few studies reported a statistically significant difference [11,31]. Moreover, it should be noted that NP score data among the mentioned studies are not uniform; Chan et al. applied the Lildholt scoring (Lildholt et al. graded the severity of NP using a 0 to 3 points system for each nostril, with a total score out of 6 [33]) for NP [4], whereas in the other studies, the calculation of the mean or the median of the NP score makes the results ambiguous and difficult to directly compare.

In addition, it is crucial to recognize the impact of sinonasal surgery on NP score. Indeed, on the one hand, it influences the baseline value in patients operated shortly before the biological treatment, thus covering the possible effect of mepolizumab on NP shrinking [5]; on the other hand, it overestimates the mepolizumab effect on NP score in patients undergoing surgery during the treatment period, thus resulting in an improper significance of NP score reduction [11].

A reduction in nasal symptoms has been described by all RLSs, except for a small case-series in which a disconnection between the effect of mepolizumab on upper and lower airways was observed [4]. Among the studies that reported favorable effects of mepolizumab on sinonasal symptoms, only a few of them measured the real improvement in terms of disease-specific QoL through the SNOT-22 questionnaire [5,11,29]. When available, a statistically significant reduction in mean SNOT-22 score compared to baseline was achieved similarly to what resulted from the present study (mean t0 SNOT-22 54.8 ± 25.9; mean t12 SNOT-22 31.5 ± 21.3, *p* < 0.0001). However, due to the limited data available, it was not possible to compare the magnitude of the clinical improvement among the identified RLSs. Interestingly, the *delta* SNOT-22 score in the present study was −22 points (IQR 31), which is almost double that obtained from the SYNAPSE study (−13.7 points) [32], to indicate a more consistent clinical effect in real life. We suppose that such a discrepancy might depend on samples’ traits, characterized by patients with severe nasal polyposis in the SYNAPSE trial and severe asthma with comorbid CRSwNP in the present study. The coexistence of the relevant clinical impact of asthma possibly turned into a greater subjective benefit than in patients mainly affected by nasal polyposis.

The clear and reproducible therapeutic effect of mepolizumab both on the clinical control of asthma, assessed by ACQ or ACT, and on the reduction in circulating blood eosinophils has been widely reported by previous studies [4,5,6,28]. A statistically significant reduction in blood eosinophils emerged also from our study (mean t0 blood eosinophils 804.7 ± 461.5 cell/µL; mean t12 blood eosinophils 107.5 ± 104.6 cell/µL, *p* < 0.0001). Indeed, in accordance with existing literature, the present study proved that 64% of patients benefited from asthma clinical control, demonstrated by an advancement of at least one step in the ACT scale. Furthermore, patients with clinically uncontrolled asthma showed at baseline both a median NP score and mean blood eosinophil count significantly higher than patients with partly controlled asthma (unpublished data). These data suggest that the baseline degree of asthma clinical control, in addition to running parallel to the extent of sinus disease (in terms of NP volume), is the only variable, among those examined, that significantly affects the baseline blood eosinophil count.

The paradox that the depletion of blood eosinophils may not proceed in parallel with the reduction in clinical and endoscopic impact of sinonasal disease has long been a subject of debate [4,10,34]. However, most of published RLSs, assessing the effect of mepolizumab treatment in SEA patients with comorbid CRSwNP, are in line with the results achieved in RCTs in terms of CRS clinical control [5,9,11,29,30,31].

The response to mepolizumab treatment can be evaluated by applying different scores and PROMs (Patient Reported Outcome Measures) in both rhinological and pneumological fields. The definition of response is arbitrarily established by the authors. As an example, one study classified *responder* and *nonresponder* patients based on clinical outcomes of upper and lower airways [30]. The present study defines responses based on nasal clinical and endoscopic outcomes. Beyond these single experiences, to our knowledge, a specific survey aimed at a comprehensive real-life multidisciplinary evaluation of mepolizumab response is still lacking. In our study, *overall responder* patients (47%) were those with a higher SNOT-22 score at baseline. Curiously, this parameter appeared to be the only positive indicator of overall response to mepolizumab, just as it is for the response to surgical therapy in CRS [17,35].

The reason allergic patients are significantly more represented among *endoscopic responders* than *nonresponders* (*p =* 0.0293) is unclear. The reduction trend observed also for SNOT-22 score and blood eosinophils in this subset of patients, though not statistically significant (unpublished data), seems to indicate that allergic patients may be more responsive to mepolizumab treatment, possibly due to either an unclear effect of the drug itself or an intrinsic feature of allergic patients.

In conclusion, studies on the efficacy of mepolizumab treatment in patients with severe asthma and CRSwNP in the real-life setting are in line with the results obtained in RCTs for symptom control and blood eosinophil count reduction. Data on NP score reduction remain controversial. More extensive and uniform outcomes in RLSs are necessary to increase the accuracy of results.

AEs in mepolizumab treatment are cited. An overall prevalence of treatment-related AEs is reported in 15% of patients; any serious events, reported in the SYNAPSE trial, occurred in 6% of treated patients even though they were not considered treatment-related [8]. In our series, the 4.4% prevalence of AEs leading to treatment discontinuation is slightly increased compared to the SYNAPSE trial (2%).

The present study shows several limitations. It is a retrospective analysis on a small and diverse sample size. Larger and more uniform series are needed to verify these results. Secondly, all patients were treated with mepolizumab and, therefore, a control group is lacking. We have shortcomings regarding biological parameters. Indeed, other biomarkers of type 2 inflammation (e.g., tissue eosinophil count, serum periostin, and exhaled and nasal nitric oxide) were not available. Furthermore, we were unable to assess the radiological changes after treatment. In fact, sinus-computed tomography (CT) scans are repeated in daily practice only in refractory cases awaiting surgery or in case of a clinical need. We also did not extensively evaluate the impact of surgery on outcomes, because the variability in the type of surgical procedure did not allow for comparisons between patients. Lastly, asthma improvement could not be verified through respiratory functional data, because these were not available for all patients.

## 5. Conclusions

Treatment of CRSwNP is currently undergoing a rapid evolution since the introduction of biological therapies targeting type 2 inflammatory pathways. After observing the promising results obtained in the management of SEA, studies focusing on the effect of these therapies in CRSwNP are in the spotlight of the scientific community. There are still many open questions on how to properly and timely integrate biologics in CRSwNP care and to conveniently select suitable patients based on a much more comprehensive phenotyping and a focused endotyping.

In this study, mepolizumab was found to significantly improve the CRSwNP outcome parameters in a population of severe asthmatic patients. An overall benefit on clinical and endoscopic aspects has been demonstrated with a reduction in symptoms severity, nasal polyps’ volume, and blood eosinophils. No clinical feature emerged to outline the profile of a “typical” responder patient, except for baseline SNOT-22 score, which seems to affect the response to treatment (the higher the t0 SNOT-22 value, the greater the response to treatment).

Further studies would be necessary to supplement these preliminary evaluations both with a more complete clinical scoring set and with other biomolecular parameters that may better reflect an inflammatory profile dominated by IL-5 and eosinophils.

## Figures and Tables

**Figure 1 jpm-12-01304-f001:**
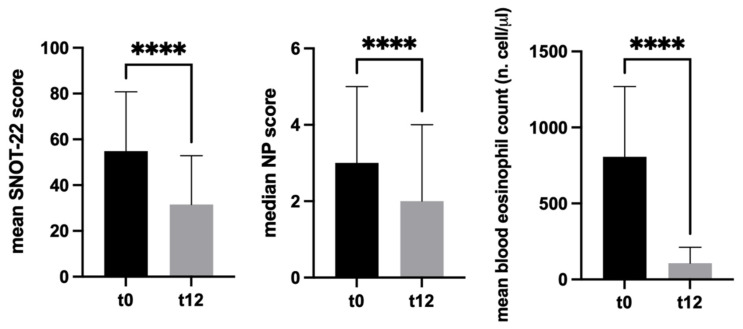
Mean SNOT-22 score, median NP score, and mean blood eosinophil count at t0 and t12. **** indicates *p* < 0.0001.

**Table 1 jpm-12-01304-t001:** Demographic and clinical data of the cohort (*n* = 43 patients).

Variables		*N*. (%)
mean age at baseline (SD)		52.2 (10.9)
sex M, F *n*. (%)		18 (42%), 25 (58%)
smoke habits *n*. (%)	smoker—ex smoker nonsmoker	16 (37%) 27 (63%)
respiratory disease onset *n*. (%)	concordant early CRS—asthma concordant late CRS—asthma discordant, early CRS—late asthma discordant, late CRS—early asthma	18 (42%) 18 (42%) 3 (7%) 4 (9%)
NSAID sensitivity *n*. (%)		15 (35%)
Seasonal and/or perennial inhalant sensitization *n*. (%)		24 (56%)
type of ongoing medical therapy	INCS spray INCS in squeeze bottle continuative OCS intermittent OCS (at least 2 courses/year) antihistamine no therapy	25 15 7 13 15 3
previous mAb therapy *n*. (%)		12 (28%)
nasal surgery *n*. (%)		32 (74%)
timing of surgery	before mAb therapy starting during mAb therapy	27/32 (84%) 5/32 (16%)
type of major surgery	polypectomy FESS ESS ESS + frontal sinusotomy DRAF type 3	4 10 14 4
median number of surgeries *n*. (IQR)		1 (3)
mean age at first surgery *n*. (SD)		39.9 (12.4)
median baseline IgE kUI/L (IQR) *		171.5 (329.2)

* Available for 30 patients. SD: standard deviation; M: male; F: female; CRS: chronic rhinosinusitis; NSAID: non-steroidal anti-inflammatory drugs; INCS: intranasal corticosteroid spray; OCS: oral corticosteroid; mAb: monoclonal antibody; FESS: functional endoscopic sinus surgery; ESS: endoscopic sinus surgery; IQR: interquartile range.

**Table 2 jpm-12-01304-t002:** Real-life studies evaluating the effects of mepolizumab on CRSwNP outcomes in severe asthmatic patients.

Study	Type	* N*. of Patients	Weeks of Follow Up	Analyzed Variables	Statistically Significant Outcomes	Results	Limits
Yilmaz et al. 2020 [28]	R	16	24	OCS, asthma exacerbation, ACT, FEV1, blood eosinophils, NAS	Asthma exacerbation ACT Blood eosinophils	The number of asthma exacerbations within 24 weeks significantly decreased and a significant increase in ACT scores was observed despite the decrease in daily OCS dosages. There was no significant difference in FEV1.	Small sample size Short-term study No control groups
Chan et al. 2020 [4]	R	6	40	Lildholt NPS, blood eosinophils, CRS exacerbations	Blood eosinophils	Patients responded favorably to mepolizumab in terms of asthma control, but their CRS disease persisted and, in some cases, continued to worsen	Absence of PROMS Small sample size No control group
Sposato et al. 2020 [9]	R	69	48 (24–53)	Subjective nasal symptoms improvement	-	In severe asthmatic patients, a greater reduction in nasal symptoms was observed in patients with nasal polyps (76%) compared to patients without (45%)	Absence of rhinologic scoring systems Not all patients were evaluated No control group
Bandi et al. 2020 [11]	P	20	52	SNOT-22, SNOT 1-12, NPS, LKS, CRS clinical control, blood eosinophils	SNOT-22 SNOT 1-12 NPS CRS clinical control	Improvement in nasal symptoms after 52 weeks of treatment, which was not associated with significant improvement in endoscopic findings	Small sample size No control groups Lack of respiratory functional data
Detoraki et al. 2021 [5]	P	44	52	SNOT-22, NPS, blood eosinophils	SNOT-22 Blood eosinophils	Significant reduction in SNOT-22 and a decrease in NPS compared to baseline. Significant decreases in blood eosinophils and mean prednisone intake were also reported	Small sample size No control groups
da Costa Martins et al. 2021 [29]	R	12	52	OCS, asthma exacerbation, SNOT-22, NCS	OCS, asthma exacerbation, SNOT-22, NCS	A reduction in asthma exacerbations and systemic corticosteroid therapy was observed. In parallel, there was also a statistically significant improvement in sinonasal symptoms evidenced by a reduction in the average total score on the Sino-Nasal Outcome Test 22 (*p* = 0.008) and Nasal Congestion scale (*p* = 0.010).	Small sample size Short-term study No control groups Endoscopic outcomes not evaluated
Meier et al. 2021 [30]	R	19	28 (4–108)	NPS, nasal symptoms	-	Treatments with mepolizumab showed the best success rates compared to other biologics; however, a correlation between biomarkers and treatment success could not be found	Absence of PROMs Small sample size No control groups Nonhomogeneous follow-up time
Tiotiu et al. 2022 [31]	R	21	24	Nasal symptoms, NPS, CRS exacerbations, CT sinus imaging, blood eosinophils	Nasal symptoms NPS CRS exacerbations Blood eosinophils	Significant improvement in nasal symptoms (except pruritus) and decrease in endoscopic score, blood eosinophil count, and number of CRS exacerbations	Absence of PROMs Small sample size Lack of baseline homogeneity No control groups Short-term study
Present study	R	43	52	SNOT-22, SNOT 1-12, SNOT-22 individual symptoms, CRS clinical control, NPS, blood eosinophils	SNOT-22 SNOT 1-12 SNOT-22 individual symptoms NPS Blood eosinophils	Significant improvement in nasal symptoms and quality of life, significant improvement in endoscopic findings.	Small sample size No control groups Lack of respiratory functional data

P: prospective study; R: retrospective study; SNOT: SinoNasal Outcome Test; NPS: Nasal polyp score; LKS: Lund–Kennedy score; CRS: chronic rhinosinusitis; CT: computed tomography; PROMs: Patient-reported outcome measures, OCS: Oral Corticosteroid, ACT: Asthma Control Test, FEV1: Forced Expiratory Volume in the 1st second.

## Data Availability

Data are available upon reasonable request.

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
