# Peer review of "Mepolizumab Improves Outcomes of Chronic Rhinosinusitis with Nasal Polyps in Severe Asthmatic Patients: A Multicentric Real-Life Study"

_jpm, 2022, doi:10.3390/jpm12081304_

Round 1
Reviewer 1 Report
The authors reported a retrospective study of mepolizumab in patients with chronic rhinosinusitis with nasal polyps in severe asthmatic patients. This study demonstrated the clinical significance of meberlizumab in the treatment of severe asthma patients with chronic rhinosinusitis and nasal polyps. The article had profound implications for asthma and nasal polyps. However, there are still some problems in the article, which I hope can be explained or corrected.
1. The author only analyzed the therapeutic effect of meberlizumab, but did not compare it with traditional treatment methods or blank control group. If these data are added, the conclusion will become more convincing
2. The article mentioned that two patients stopped treatment because of serious adverse reactions. Were these adverse reactions related to meberizumab?
3. Were these patients still taking corticosteroids or other drugs during treatment?
4. The author mainly recorded the changes of relevant indicators before treatment and 1 year after omalizumab treatment, and the interval was long. Is there any relevant data after 3 months and 6 months of treatment? Multiple group comparisons may be helpful to studying the curative effect of meberlizumab in treating patients with chronic rhinosinusitis with nasal polyps in severe asthmatic.
Author Response
Dear reviewer,
We do appreciate your comments and we hope you would accept our reply.
(1) As reported in methods session, this is a retrospective observational study in which we revised available cases of patients affected by CRwNP and treated with mepolizumab for controlling severe asthma. As already expressed in the discussion, we are aware that the study has the limit of not having a “control group” of patients with severe asthma and CRSwNP treated with the standard maximal medical therapy, but these data were not available. Further studies incorporating also these analyses would add, of course, much more value to results. However, we must say that the objective of the study was to verify the effect of mepolizumb on sinonasal aspects in patients with severe asthma and not to compare the effects of treatments available for CRSwNP. In this sense, the employment of biologics in asthma is already considered a traditional therapy since biologics are officially integrated in step 5 asthma treatment.
(2) We mentioned that 2 patients stopped treatment because of adverse events. One patient experienced gastrointestinal disorder which increased nearby the administration of the drug and not disappeared. The same was for the other patient who started with simple arthralgia and worsened up to evidence of signs of arthritis. Symptoms were so invalidating that pulmonologists decided to suspend treatment. Both patients had progressively benefit from suspension and symptoms resolved within some months.
(3) Therapies taken during the observation period are reported in table 1. These include steroid nasal therapies, which remained unchanged in all patients until the end of the study. The same was for OCS intake. It was not mentioned in the manuscript, because taken for granted, that patients were of course under standard inhalant therapy also for asthma.
(4) The decision to record changes between baseline and after 12 months was dictated by the fact that uniform measurements were available only for this interval. We do agree that it would have been interesting to evaluate therapy effects in shorter timelapse to give information also on the timing of the effects. Prospective observational studies would be useful also in this sense.
Reviewer 2 Report
This study seems quite well performed, yet I assume if it hadn't been conducted retrospectively, the quality of your work could be much better.
Author Response
Dear reviewer,
We do appreciate your comments and we hope you would accept our reply.
We are aware of the limitations of the study. You should consider that Mepolizumab is not yet available to be used by rhinologists in Italy. Therefore, the idea of the study was to have a preliminary view on the effects of Mepolizumab on sinonasal aspects of CRS with nasal polyps and we took advantage from pre-existing pulmonologist's databases.
Reviewer 3 Report
This is the study about determining the efficacy and safety of Mepolizumab for CRSwNP in severe asthmatic patient. The study provides novel knowledge especially the predictive factors for determining responder to Mepolizumab in CRSwNP parameter. The authors clearly defined the objective of study which provide additional knowledge to the existing (ref#9) that analyzed CRSwNP as its secondary endpoint by using patient report outcome.
There are some comments which will improved the quality of manuscript and make it easier to the reader.
1. In the abstract, the author should show the margin of score difference (pre-post comparison), esp the finding in Line#225-226 and the key finding from Figure1.
2. In the line#129-130, the classification of EPOS step of improvement should be explained in more detail, which will make the reader understand it easier.
3. In line#250, the statistical method (eg. multivariate logistic regression method) of determining predictive factor of responder (clinical, endoscopic, and both) should be displayed, especially in 'table format'.
4. The sequence of discussion (line265-381) should start from: the finding of primary outcome (margin of improvement), and then followed by the secondary outcome (determination of predictive factor for discrimination of responder vs non-responder).
In conclusion, this is the good manuscript that suitable to be published in JPM. The above suggestion/comments are meant for your consideration to make this manuscript easier and clearer for the reader.
Author Response
Dear reviewer,
we do appreciate your comments and thank for suggestions.
(1) As requested, we add the margin of score difference and the key findings in the abstract in Line#51-53
(2) We explained more in detail what the EPOS scale of disease control is for a better comprehension of the results of the manuscript (see Line#141-145)
(3) Since we employed a comparative analysis between responder and nonresponder patients looking for not casual recurring characteristics, we changed the previous sentence “Baseline SNOT-22 score positively predicted the response to treatment (p 0.0011)” in “Mepolizumab responder patients presented a t0 SNOT-22 significantly higher than nonresponders (p 0.0011)” (see Line#55-56).
(4) We modified the logical sequence of the discussion (Line#304-383) starting with the primary objectives (NPS, SNOT-22 and peripheral eosinophilia), adding margins of improvement to each of them. We subsequently discussed the overall response to treatment including other variables, beyond primary objectives, and their interrelation.